# A Multi-Stakeholder Evaluation of a Walking Football Group for People with Dementia Developed in Partnership with a Premier League Club

**DOI:** 10.3390/sports13060172

**Published:** 2025-05-30

**Authors:** Marie Poole, Alison Killen, Louise Robinson

**Affiliations:** 1Population Health Sciences Institute, Newcastle University, Newcastle upon Tyne NE4 5PL, UK; a.l.robinson@newcastle.ac.uk; 2NIHR Newcastle Biomedical Research Centre, Translational and Clinical Research Institute, Newcastle University, Newcastle upon Tyne NE4 5PL, UK; alison.killen@newcastle.ac.uk

**Keywords:** dementia, dementia support, cognitive impairment, walking football, evaluation, older adults, qualitative research

## Abstract

Dementia-friendly walking football offers a way of helping people who are less likely to engage with traditional services to participate in physical activity and support their physical, mental and social wellbeing. This addresses a gap in the current provision of post-diagnostic dementia support in the UK. However, there is a lack of evidence around such models of service provision. Through the evaluation of a newly established dementia-friendly walking football initiative provided by a Premier League charitable foundation, we explored the social, physical and mental benefits of dementia-friendly walking football for older males from underserved communities. Using a qualitative, exploratory approach, we held focus groups with service providers and service users to understand their views and experiences of delivering and attending dementia-friendly walking football sessions. A thematic analysis of the focus groups revealed three main themes relating to the importance of football to cultural and individual identity, namely, ‘for the love of the game’, ‘team players’ and ‘a game changer’. We also highlight how a multi-disciplinary, research-based approach to the evaluation of a service and identification of service improvements can both involve and benefit people living with dementia and their families.

## 1. Introduction

Dementia is increasingly recognised as a growing global public health concern, with over 57 million people living with dementia worldwide in 2021 and nearly 10 million new cases each year [1]. Physical inactivity, social isolation and depression are recognised as significant modifiable factors which can increase dementia risk [1,2]. Furthermore, physical inactivity is thought to account for around 3.8% of dementia cases worldwide, and there is growing awareness of the importance of physical activity to support and maintain cognitive capacity [3]. Physical activity for people living with dementia promotes positive feelings, provides meaningful activity and can take place despite the barriers caused by reliance on others to access the activity and the challenges caused by changes within the brain [4]. Although physical activity benefits functional, physical and cognitive health [5,6], people living with dementia are less likely to be physically active than older adults without cognitive impairment [7]. Typical factors which impact, positively and negatively, the ability of people living with dementia to engage include the provision of structured activity, support from others to participate and cognitive and mobility issues [8]. Walking football (soccer) is one way to promote access to physical activity for this subgroup by engaging or reconnecting them with the game, but there is limited empirical research in this area.

Walking football is the United Kingdom’s fastest growing sport and is considered an authentic form of football [9,10]. It was initially developed to enable men over fifty to stay engaged for longer or reconnect with playing safely by offering a reduced-intensity, non-contact game which decreased the risk of injury [11]. In the context of healthy ageing, walking football offers many benefits, including the physical advantages of moderate exercise and the benefits of socialisation [12], as people approach retirement, a time when a reduction in social contacts can lead to social isolation [13].

Walking football is promoted as an inclusive sport and has been adapted for a range of conditions, including Parkinson’s disease [14] and brain injury [15]. Walking football for people living with dementia is an emergent area, delivered via a range of organisations in the UK and beyond, but with limited evidence regarding any effectiveness as an intervention.

The number of people living with dementia worldwide is predicted to rise from 46 to >131 million between 2015 and 2050 [16]. In England and Wales, numbers are predicted to increase by 57% from 0.77 million in 2016 to 1.2 million in 2040 [17]. Thus, it is imperative to find new ways to support people and new providers to deliver effective, affordable and accessible activities. Sex differences for dementia show that males have a higher overall mid-life dementia risk. They have a lower late-life Alzheimer’s disease risk [18] but are more likely than women to develop Lewy body dementia [19]. Men in the UK, especially those from non-white ethnic and low socio-economic backgrounds, need greater support in accessing care [20] due to greater impact from the cost-of-living crisis [21].

In the context of football-related dementia support initiatives, the relevance of such provision for men is often emphasised. Using football to engage men living with dementia who are reluctant to engage with more talk-based traditional support services such as dementia cafes [22] is increasingly recognised as an effective approach [23,24]. Watson et al. highlight the connection between gender, age and sport, suggesting that older males are often ‘deeply wedded to the socio-history of British sporting culture’, making initiatives such as Sporting Memories (sports-based reminiscence and activities) more appealing to them than to females of similar ages [25]. In the UK, the majority of adult males will have played football at school and in their youth, and many will have regularly watched professional games and had footballing role models. Football was not an option for girls in school sports and between 1921 and 1972; women were banned from playing in Football League grounds in the UK [26]. Thus, the elements of nostalgia and reminiscence are likely to have greater significance for men with dementia than for women. Women who play walking football have been understudied [27], and whilst women-only walking football teams exist [28], these are uncommon, suggesting that women with dementia who play walking football are likely to be very much in the minority.

A small number of evaluations of walking football initiatives for people living with dementia have identified benefits and barriers. MacRae et al. [29] focused on the social impact alongside best practice recommendations, whilst Carone et al. [30] explored the impact of a physical activity- and sports-based (including but not exclusively football) initiative for people living with young-onset dementia and their families through a football club foundation. Recommendations included the need to modify general walking football programmes to increase accessibility for participants living with dementia [29].

The multi-dimensional potential benefits of walking football for dementia highlighted within male populations include social wellbeing in the form of connectedness, teamwork, a sense of responsibility to peers, confidence and the development of new skills [31]. Moreover, the importance of ‘banter’, ‘camaraderie’ and forging friendships on and off the pitch are cited as significant benefits for mental wellbeing in older males. These provide distraction from mental health conditions, combat isolation and loneliness and re-create previous connections and a sense of self which often centred around playing or watching sports [32]. As well as relational connections, a sense of identity can be supported through ‘embodied memories’, bringing participants a sense of pride [29] drawing on reminiscence through physical activities designed to stimulate their memories of sport and physical activity. However, stereotypes and assumptions about dementia can result in disengagement from sporting activity [33].

There are an expanding number of community football-based initiatives offering alternatives to traditional services in terms of activities and providers for people with dementia [34]. This is significant because services provided through statutory health and social care services are increasingly financially stretched [35], stimulating a drive to build links with alternative partners. These include hospices [36], community organisations [37], homecare organisations [38], Primary Care Networks [39], national dementia charities such as Alzheimer’s Scotland and Age UK [40], football associations [41,42], clubs and their associated charitable foundations [43] and local authorities [44]. Many are joint collaborations between multiple organisations [45,46]. In some cases, motivated individuals living with dementia have even developed their own groups [47].

The aim of this study is to explore the social, physical and mental benefits of a dementia-friendly walking football intervention for older males and underserved communities using focus group data from participants and the staff delivering the sessions. We also highlight the benefits of a multi-disciplinary, research-based approach to evaluating the provision of post-diagnostic dementia support.

## 2. Materials and Methods

The design employed was an exploratory qualitative study [48] using health service research evaluation methods [49]. Researchers MP and AK were embedded in the service from the beginning and adopted a ‘Go along’ method [33]. Actively participating in sessions as a shared experience created an understanding of service users and providers through informal conversations and the opportunity to appraise the cognitive and physical capabilities of the service users over time. A description of the intervention and attendees is provided in Figure 1.

Two focus groups were held to explore experiences of either delivering or attending a new dementia-friendly walking football programme at the Newcastle United Foundation and identify good practice and areas for improvement. Written consent was obtained, with all participants considered to have the capacity to consent to participate in research in accordance with the Mental Capacity Act 2005, Code of Practice [50].

Focus group one was held in May 2024 with staff from the Foundation Health and Wellbeing Team who deliver walking football programmes (*n* = 4). Three males and one female attended. Focus group two was held in September 2024 with service users (*n* = 6). All walking football attendees were invited verbally to participate and given written or verbal reminders as they preferred. Five males took part, of whom three had a confirmed dementia diagnosis, including dementia with Lewy bodies, Parkinson’s disease dementia and Alzheimer’s disease, and two support a brother living with dementia. One female carer participated with her husband.

Both focus groups were conducted at Newcastle University to foster a sense of neutrality and encourage staff and service users to be open and honest in their reflections. They lasted between 64 and 75 min with time before and after to brief and debrief. Participants were not financially reimbursed, but lunch and refreshments were provided. Recordings were transcribed by a university-approved transcription company before being checked and anonymised by the authors.

A thematic analysis approach was applied to the data [51]. Transcripts were read by both researchers, who made independent notes, including verbatim quotations, and compared these for convergence and divergence. For both focus groups, the two researchers identified similar themes. These were further discussed to confirm that their understanding of each theme was coherent. Football-themed titles were given to each theme. A separate report of the findings of each focus group was produced and shared with the foundation management, the delivery team and the service users. The researchers also collaborated with an illustrator [52] with experience in dementia research to develop visual interpretations of the anonymised data with the intention of making the service user report more engaging and accessible.

Five key themes were derived from the service user focus group, whilst seven key themes were identified from the service provider group (see Table 1). Considerable overlap was identified, which resulted in the following three overarching themes: Theme 1. For the love of the game; Theme 2. Team players; Theme 3. A game changer. We have used famous football quotes to contextualise the meaning of our themes. These are described below, accompanied by anonymised quotations.

## 3. Results

### 3.1. Theme 1. For the Love of the Game


*“Football is more than a game; it is a passion. It is something that comes from inside and touches people’s hearts.”—Yaya Touré*


The culture of football and the connections people feel to their individual clubs and charitable foundations relate to a strong sense of identity for both individuals and communities. This creates a sense of belonging and is an important part of a sense of self. In relation to experiencing life with dementia, initiatives such as walking football provide a connection and a constant in the context of a condition characterised by loss. This can include a loss of memories, loss of mental and physical capabilities, loss of social connections and loss of self and identity. One participant expressed the centrality of football to her husband’s life and how attending walking football gives him a sense of connectedness to his past whilst also fostering a sense of purpose in his life:

“Football’s his thing, isn’t it? So that’s that. And just a purpose on a Wednesday, we know what he’s doing and it’s his thing. Now he doesn’t do things on his own anymore, like he would do normally, so it’s just like it’s his thing to do.”(Name 9)

The support of the staff team who share his passion for football has had a positive effect on his ability to communicate.

“About a month ago [participant] spoke for the first time in a group setting, and his wife couldn’t believe it. She actually said, “I can’t believe you’ve managed to get him to talk.” But I think that’s the comfort of football again. He understands football.”(SP03)

One of the staff team remarked on how the club’s football history provides a framework for another participant to structure his memories and relate to others through a shared understanding of games. Whilst supporting recognition and recall for this individual, it also enables his connectedness with others. Walking football has enabled him to share these memories with like-minded people:

“And what’s beautiful about it, there’s a participant as well called (Name 6), and he actually uses football, particularly Newcastle United, as a mechanism to, basically, recognise time. And what that does-So, for example, if you went back to a cup final in the 70s, for example, he’ll be able to then think of life events that occurred in and around, and it helps him build up his brain again, they’re his words. So it’s not just football, but predominantly we’re using the power of Newcastle United and football to go back to events which can trigger other things too, which clearly works, which is phenomenal.”(SP03)

Furthermore, at the community level, the staff team recognised that football has a unique way of encouraging people living with dementia to tap into memories and maintain existing skills. They recognised that the club and foundation can play an important role in dementia support. This was expressed in terms of general security and trust in the club. People had a shared understanding of the power of the club badge; they knew what to expect, and it was predictable. The team also identified that the club could be seen as a safe space in comparison with other unfamiliar providers who offered dementia services in the community.

“About people going to, perhaps, NHS (National Health Service) services, they may not do it for whatever reason, but they see this, and they want to do it because it’s their comfort blanket, the power of Newcastle United…”(SP03)

The continuity in the life of the person living with dementia which the club represents also extends to a sense of belonging to the regional identity that is strongly intertwined with the club and its history in the community. Those attending the sessions, men from predominantly lower socio-economic groups, had a shared masculinity rooted in childhood experiences of attending matches with dads, uncles and brothers along with their peers.

“It’s a constant in their life, isn’t it? Whether they support football or whether they’re interested in football, they know. If you live in the North East you know, do you know what I mean? So it’s the safety of being behind the badge, isn’t it?”(SP02)

An important factor in participating in walking football for several participants was being able to reconnect with the physical game, which they may have had little recent opportunity to do and have missed.

“It’s been a life changer, probably, because I didn’t think- I’ve got new knees and hips and everything, and I didn’t think that would take part in any sort of football now. I really missed the football. And when you gave me the opportunity to come the walking football, I didn’t think that I would be taking part in anything. And I think (Name 5) was the same. But then it just turned around, everything.”(Name 7)

The football game followed familiar rules whilst being modified to be safe and inclusive. Playing as part of a team according to specific gameplay helped participants alleviate some of the losses experienced due to their cognitive impairment, as one participant identified.

“… you’re using your mind as well as your feet, thinking about where you should be and… So it’s not just the actual kicking the ball, but it’s trying to use your head, your positioning and one thing or another.”(Name 2)

The team identified that attending walking football practices could be the ‘carrot’ to motivate people to work on social skills which were now under-used and in danger of being lost. They felt more traditional dementia support services may struggle to have that ‘pull’ within the local community, which includes areas facing social disadvantage.

“We’ve got a participant who said to me last week he doesn’t use any form of public transport up until the start of the sessions he’d been coming to, and he uses that one session as his time out. So he gets public transport to the session, that’s the only time he uses it throughout the week. But that’s giving him independence and confidence that if he needs to do that, he knows he can… he didn’t want to have to rely on someone to bring him, because then what if they couldn’t? So this spurred him on to learn this route“.(SP04)

The majority of attendees at dementia-friendly walking football are keen football fans. Football has and continues to be an important part of their current lives and memories. The social aspect of the sessions helps them to reconnect with their identity as fans who can gather with other fans and appreciate their mutual love for the club. Alongside the beneficial aspect of social engagement, there are also elements of purpose and routine and a strong motivation to attend and stay involved.

“Coming to the walking football is an important part of the week. It puts a stake in the ground on a Wednesday for me. Otherwise, I could see myself quite simply isolating more and more. It’s fascinating, the enthusiasm and the passion, and it rubs off. So I report back each week, after I’ve been, to my sons, who are around the world. ‘I’ve met (Name 8) and I’ve got involved and we’ve predicted this and predicted that’.”(Name 4)

Sessions include a social element in the form of either a structured group discussion or informal opportunity to chat. Some participants highlighted how this discussion time is different from other dementia groups they attend by focusing on football or pleasurable reminiscence such as music or pets rather than on dementia. They described enjoying these topics and considered them inclusive and relevant to people from a range of backgrounds and experiences.

“You get to know people better, don’t you, when you know what they like and what they dislike. It’s just good to hear everybody’s opinion.”(Name 5)

Although dementia was never the focus of the discussion, this was sometimes spontaneously brought up by participants. This approach was valued by attendees and provided a relaxed environment perceived as a social rather than a therapeutic space despite the clear therapeutic benefits.

For the staff team who had little to no experience with dementia, the impact was surprising and highly gratifying. They shared their observations of the benefits. These included participants using their memory, practicing talking through sharing about past events, enjoyment, growing in confidence and ‘coming out of their shell’, making new connections with others in the group, bonding with the staff and researchers, becoming more relaxed and less inhibited, having physical exercise and having somewhere to feel safe and included. The focus group took place just six months after the start of the group, and one remarked on the impact it had had on participants in such a short time.

“I didn’t really know what to expect in the beginning and now it’s just got- It’s blown every expectation that I even subconsciously had, out of water, in what it’s doing for these participants, literally.”(SP02)

### 3.2. Theme 2. Team Players


*“I am constantly being asked about individuals. The only way to win is as a team.”—Pelé*


Developing a walking football initiative for people living with dementia requires a multi-disciplinary approach, with the foundation team at the forefront, bringing in their mix of specialist skills and knowledge of football and walking football sessions delivered with other groups. They also utilised support from a colleague running dementia walking football sessions elsewhere with whom they could discuss the various components involved in delivering a session. The team was skilled at tailoring both the social and physical aspects of the sessions to the abilities of the individuals attending, which was highly effective.

“What I like particularly, and I had to get my head around it to start off with, was the support that the staff give to those who are not too able and not so mobile and, maybe, not directionally very functioning very well, to help them hit the ball and score a goal.”(Name 4)

However, although some of the team had personal experiences with dementia through family, none had formal expertise in supporting people living with dementia. Embedding dementia researchers and a dementia support worker from a local charity within the walking football service provided emotional and practical support and dementia psychoeducation for both staff and participants. It was a novel way to facilitate the ability of an organisation without dementia expertise to provide this activity.

The foundation staff highlighted the benefits of working in collaboration with other agencies who could bring their knowledge and experience of dementia to the sessions. Prior to the first session, some staff had attended a half-day training workshop on dementia delivered by one of the researchers. The staff emphasised how much they had valued attending this course, which they felt had increased their knowledge and confidence by being able to draw on practical tips which they considered easy to put into practice. The team reflected that they had put many of these strategies in place.

“Just to echo what everyone said already about the training, that was phenomenal. It was really relatable and really- There were practical elements in there, which you can really learn from straightaway and utilise, and a lot of that I’ve already started to put into my own practices.”(SP03)

In addition to attending a one-off training session, the team also identified the importance of ongoing weekly support to deliver their sessions, giving them confidence and prompting awareness of potential difficulties. This included recognising forms of spontaneous feedback such as smiling and increased confidence and identifying what aspects of the provision have led to that. It also involved guidance around the fluctuating nature of some types of dementia and accommodating breaks or closer guidance for some players or understanding aspects of some dementia types which could impact safety on the pitch. As a result, the staff team was increased to facilitate 1:1 support for players with poor balance or visual disturbances to reduce the risk of falls.

“He tried to kick the ball once and he looked like he was going to fall over. And I just immediately recognised that situation, so I just was close to him all the time so if he did- If he looked like he was going to fall over I was always there.”(SP01)

The majority of attendees were either prompted by a family member or more commonly encouraged during a clinical appointment or by a dementia charity worker to attend the sessions. It is not necessarily the case that if an activity is available, people with dementia will join. Most participants needed multiple prompts to attend their first session, although having experienced the group, they were enthusiastic about returning. There are often misconceptions about activities for people with dementia, and many activities conform to a stereotype which may be welcomed by some but may not suit the demographic drawn to walking football. This reinforces the need to widen the range of opportunities to improve the equity of access to support. Initial attendance can be a challenging barrier to overcome and may require prompts from multiple partners, including discussions at clinical appointments, advertisements within the club and matchday media.

“You’d (Researcher 2) tried to encourage us a few times, and he wouldn’t come. But then when he eventually has come, he does enjoy it. He likes the football side of it.”(Name 9)

This highlights the benefits of focusing a dementia-related activity around an interest and organisation that a potential attendee views as credible and strongly identifies with. It may also reflect the reluctance of many older men to join groups. Many men of this generation within the area served by the club have had working lives involving long and tiring hours, so may have little previous experience of joining a group for a social activity.

“Never heard from my brother for ages, then all of a sudden, he got in touch and said, “Do you not fancy coming along to the walking football and the memory café?” I said, “Never heard of them.” He said, “I’ll come and pick you up.” And I thought, “I’ll go the once,” and that would be it for me.”(Name 5)

In just a few weeks, an unprompted and noticeable bond developed between the participants. This included positive comments for effort regardless of the result of the game and passing to players to keep them involved regardless of their skills. People also recognised and welcomed other participants when they arrived, although rarely by name.

“It’s that camaraderie between them all, because when ones struggling in the group, say with the talking, all the time they then help each other, don’t they? I just love it. I think they’ve really come together as a group, which, I suppose, I’d never expected. I didn’t expect that. I expected them to come, but as individuals. But now they’ve become a sort of team, haven’t they?”(SP02)

Some participants had not been inspired to attend other dementia support services, but others were unaware of what was available to them. Attending walking football at a credible and trusted venue and speaking to the researchers and support workers was a first step in accessing further support. Care partners who had often identified their need for additional services but found the person with dementia unwilling to agree were also ideally placed to have these conversations whilst their family member was playing football. Partnering with a range of agencies such as local ageing, dementia and carer charities and health and social care services to promote the group and encourage recruitment is important to maintain sustainability because people become less able to attend over time due to disease progression.

### 3.3. Theme 3. A Game Changer


*“My game is based on improvisation. Often a forward does not know what he will do until he sees the defenders’ reactions.”—Ronaldinho*


The team appreciated the particular need to be responsive to participants’ engagement and flexibly adapt the sessions to enhance their experience and participation. They embraced making changes to the sessions to maximise positive impact for the group and individuals. This applied to the discussion activities as well as playing walking football, for example, learning to recognise individual needs, such as that one person rarely wanted to speak, whilst another needed time to express themselves and benefited from the initial questions being directed towards others.

“Last week, the discussion was so good it had to carry on, because it was flowing, people were talking, they were all engaging, they were all interested and were all listening. So it had to go on because if you’d have stopped it just to go play the walking football, it wouldn’t have had the same impact.”(SP01)

As well as being able to responsively calibrate the sessions in the moment, the team was also keen to think about some of the challenges around keeping the sessions engaging and inclusive:

“It’s good to reflect on the challenges as well, and how we can improve this session. I think we’ve probably touched on what the challenges are. Maybe thinking of what the talking points could be during the session, because, like you said, we’ve got people that are interested in the football and maybe some that aren’t. But I think we’ve put things in place that can overcome that.”(SP04)

Although not originally envisaged by the delivery team, the value of evaluation and co-creation as key features to enhance the provision of dementia-friendly walking football was recognised and embraced by the staff team. They appreciated that working with the attendees to find out their suggestions and recommendations to improve their service could mitigate some of the practical challenges impacting their ability to engage in the sessions:

“The other thing that I did bring up with you was, (name 6) wanted, maybe, to wear names …In big writing, so not our ones where you’d have to squint and look up but just so that he could remember.”(SP02)

Where changes were introduced in response to participants’ comments, this was noted and appreciated. For example, the discussion was shortened with less expectation for members to contribute.

“Things have changed a bit now or maybe I’ve changed a bit as well, so I’m more outspoken. But I certainly wasn’t then. So it’s worked, so I’m happier with it, and the football is a relaxation. Even though I’m still no good at football.”(Name 2)

Collaborating with the research team enabled both service providers and service users to reflect on the benefits of providing and attending dementia-friendly walking football and determine areas for opportunity for improvement. Creating this space to address some of the issues and challenges led to identifying a range of suggestions and adaptations.

“You learn from doing, but it’s important you reflect, like this now. You can’t just plan and do, you’ve got to reflect to understand what’s not working and what is working, but mainly what’s not working or what can be improved.”(SP03)

For the researchers, reflection on the delivery of the sessions went beyond a research activity and became embedded in the practices of the team and the delivery of the sessions.

## 4. Discussion

This research builds on previous research on men with young-onset dementia in accessing walking football [30] by exploring the experiences of both the staff delivery team and participants at a range of stages of dementia and different ages attending dementia walking football in association with a football club foundation in the northeast of England.

Football clubs have high credibility for potential service users who often have lifelong affiliations. Dementia-friendly walking football delivered through an institution perceived as trustworthy, safe and friendly offers a unique way for people living with this condition to access much-needed physical, social and psychological support. In the context of limited and underfunded post-diagnostic dementia support services in the UK [53] and more broadly [54], novel providers are likely to become ever more critical in enabling people living with dementia and their families to access support which meets their needs [55]. The evaluation of this service highlighted how attendees gained increased confidence and maintained or improved their physical, cognitive and social skills.

Furthermore, non-healthcare organisations such as football clubs, which are culturally integral to many people living with dementia, particularly those living in socio-economically disadvantaged communities and older males [21], can deliver dementia provision to people reluctant to access services and at increased risk of isolation. Social isolation is modifiable, and creative programmes and interventions can play an important role [56]. These populations may be persuaded to engage by the ‘power of the badge’ [30] of their beloved team or by connection to the game itself.

Attendees valued the lack of focus on dementia during the football sessions, which contrasted with their perceptions or experiences of other provisions. However, this environment enabled participants no longer able to attend matches or visit pubs to watch football screenings to identify as individual and team footballers and football supporters. This helped them to maintain their identity, cultural connections and sense of self, core aspects of dementia support. Other interventions promoting this sense of self are typically arts programmes, either reminiscence or therapy based [57].

The physical benefits of walking football were identified by both attendees and staff, yet information or advice about dementia-inclusive or dementia-specific physical activities is not routinely provided around the time of diagnosis [58]. Furthermore, when such information is given, it rarely includes dementia-friendly walking football. This is despite frequent national policy recommendations highlighting the risk of not following recommended exercise guidelines regarding increasing cognitive problems [59] and government guidance around secondary prevention (preventing complications and worsening of conditions) highlighted in the major conditions strategy [60], alongside people living with dementia being denied opportunities to maintain or improve their physical health [58]. National dementia charities such as the Alzheimer’s Society [5] and Dementia UK also champion the positive role of sport and exercise for people living with dementia and their carers [61], both as participants and spectators involved with sports [62], and Dementia UK has a designated Consultant Admiral Nurse for Sports and Dementia. This underscores the significance of partnership in working to ensure that sporting initiatives are promoted through clinical and community services. A repository of dementia-friendly walking football programmes could be a valuable asset to enable health, social and voluntary care service providers to signpost people.

From the perspective of the Newcastle United Foundation, the opportunity for a service evaluation as used in health services [63] allowed them to verify the impact of their service using appropriate and robust methodologies and work towards their quest to be ‘game changers’. As evidenced in this research and other studies [30,64], working in collaboration with researchers with experience in this field is important. Similar services for people living with dementia delivered outside healthcare settings by staff other than healthcare practitioners may also benefit from analysis using resources originally designed to support NHS service evaluation [65] to build a robust evidence base regarding the progress or success of their initiatives. This could also alert providers to the potential risks of delivering a service without a partnership with a provider of dementia service ‘expertise’ such as an NHS Trust [30] or dementia charity, such as a lack of dementia knowledge in the staff team, lack of understanding of safeguarding people with impaired capacity and lack of awareness of where to signpost people for further support.

We recognise that the findings of this research are limited to the views and experiences of a small number of service users and providers. This is reflective of the relatively small numbers of service users who attended at that time, although the whole staff team delivering the service participated in the focus group. However, since then, the number of service users has since grown considerably, and the team has expanded. As such, we would like to engage with larger numbers of participants in the future to ensure that a broader range of perspectives and experiences are considered, facilitating ongoing reflection and change.

This research was conducted in the UK, and although further research is needed to build a comprehensive evidence base across a range of geographical settings and diverse characteristics (for example, all people living with dementia were white British males), the findings have relevance to a broader European and global context. There is a growing global awareness of the benefits of walking football for older people [66]. Dementia-specific walking football initiatives appear to be growing based on social media posts from beyond the UK, including Australia [67], although there is currently no empirical evidence on current initiatives. The not-for-profit European Football for Development Network (EFDN) [68] website previously hosted information about dementia-friendly football initiatives in Europe but no longer does so. As such, there is a dearth of empirical evidence about how such initiatives function and their benefits. Further research is needed regarding the inclusivity of older people, including those living with dementia, in sport and walking football [69].

Existing guidelines for modifying walking football for people living with dementia [29] are a useful and practical resource. However, through the current service evaluation, a range of recommendations were made by staff and attendees which could further enhance this provision (Figure 2). These adaptations considered the needs of participants with different types of dementia. For example, people with Parkinson’s dementia may become very stiff when sitting and experience difficulty in starting to move again. They can have sudden drops in their blood pressure when they stand and have problems with balance and visual perception. Others have difficulty in concentrating for long periods or have hearing deficits which can be exacerbated by the distractions and echoes of a larger space such as a sports hall.

## 5. Conclusions

Exploring a walking football programme for people with dementia through a focused discussion has helped to confirm what is working well and to identify possible areas for future development. The service has had a highly positive start and grown well over a short time. Contributing to this are the trust and faith people have in the club brand and the drive and enthusiasm of the foundation team. Loyalty to the club and the close relationship with the local community held by the foundation as its charity arm means this offer is far more effective than more usual forms of dementia support such as dementia cafes and generic activity groups. The location of the foundation building in a lower socio-economic area of the city and the significant interest in football from local men means a demographic typically unwilling to access dementia support is choosing to access this service. This illustrates the role new partners such as football clubs can play in improving inclusivity and reducing inequalities in access to dementia support.

## Figures and Tables

**Figure 1 sports-13-00172-f001:**
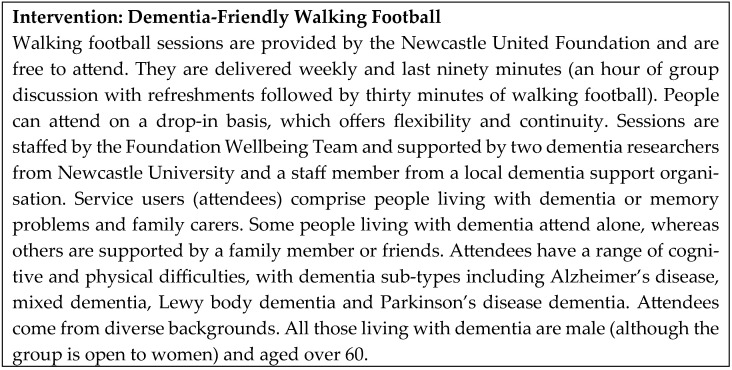
Description of the intervention and attendees.

**Figure 2 sports-13-00172-f002:**
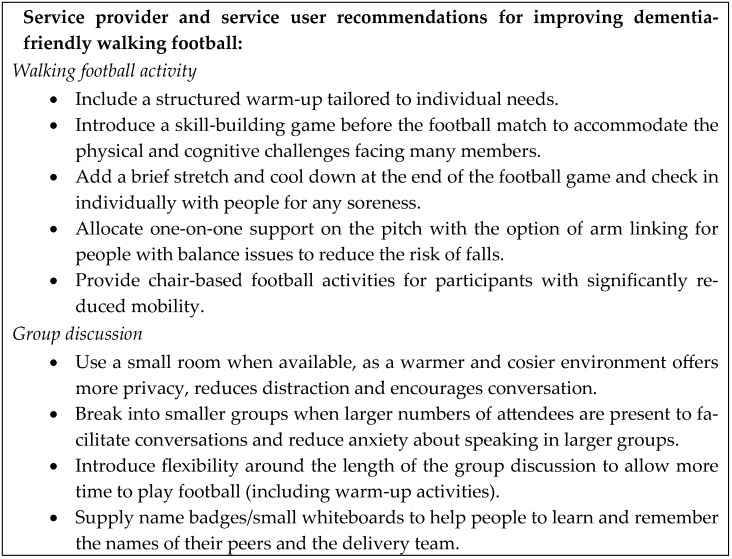
Recommendations for improving dementia-friendly walking football.

**Table 1 sports-13-00172-t001:** Themes derived from analysis of the focus groups.

Focus Group	Initial Themes	Overarching Themes
Service users	More than just a game!—Meeting the need for dementia support	Theme 1. For the love of the game
	Match ready—Motivation and prompts to attend	Theme 1. For the love of the game
	Local heroes—Foundation staff	Theme 2. Team players
	Team talk—Benefits and challenges of themed discussions	Theme 3. A game changer
	Game plan—Identifying challenges and finding solutions	Theme 3. A game changer
Service providers	United—Meeting the need for dementia support	Theme 1. For the love of the game
	Supporters’ club—Positive interactions for people living with dementia	Theme 1. For the love of the game
	Local heroes—Personal impact	Theme 1. For the love of the game
	Team players—Experience and knowledge of dementia and team skills; benefits to staff team	Theme 2. Team players
	Team tactics—Planning the sessions	Theme 2. Team players
	A game of two halves—Progression and adaptations	Theme 3. A game changer
	Next season—Future plans	Theme 3. A game changer

## Data Availability

The data presented in this study are available on reasonable request from the corresponding author due to privacy reasons.

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
