# Peer review of "A Multi-Stakeholder Evaluation of a Walking Football Group for People with Dementia Developed in Partnership with a Premier League Club"

_sports, 2025, doi:10.3390/sports13060172_

Round 1
Reviewer 1 Report
Comments and Suggestions for Authors
A Multi-Stakeholder Evaluation of a Walking Football Group for People with Dementia, Developed in Partnership with a Premier League Club
This study explores the implementation of a dementia-friendly walking football initiative, designed to engage individuals who are typically less likely to participate in conventional services. The program aims to promote physical, mental, and social well-being among people living with dementia. Notably, there remains a significant lack of empirical evidence surrounding such innovative models of service provision.
Employing a qualitative, exploratory methodology, the researchers conducted focus groups with both service providers and service users, with the objective of capturing their perspectives and lived experiences regarding the delivery and participation in dementia-friendly walking football sessions. A thematic analysis of the data generated from these focus groups identified three principal themes pertaining to the role of football in shaping both cultural and personal identity: ‘for the love of the game’, ‘team players’, and ‘to be the best’.
This study makes a valuable and timely contribution to an under-researched area, particularly given the considerable challenges associated with promoting physical activity among individuals living with dementia. The proposed intervention of walking football is both creative and socially meaningful.
However, there are two substantive points of critique that should be emphasized and presented more clearly.
First, the introduction would benefit from further elaboration on the magnitude and gravity of the issue being addressed. Specifically, it would be helpful to foreground the scale of dementia as a global public health concern:
Currently, over 55 million people worldwide live with dementia, with more than 60% residing in low- and middle-income countries. Each year, approximately 10 million new cases are identified. Dementia is caused by a range of diseases and injuries affecting the brain, with Alzheimer’s disease representing the most prevalent form, responsible for 60–70% of cases. Dementia is now the seventh leading cause of death globally and a major cause of disability and dependency among older adults. In economic terms, the global cost of dementia was estimated at $1.3 trillion in 2019, with 50% of this burden attributable to informal care-giving, typically provided by family members who dedicate, on average, five hours per day to care and supervision (WHO, 2023).
In addition, physical inactivity is recognized as a significant modifiable risk factor for dementia. It has been estimated that approximately 3.8% of global dementia cases are attributable to insufficient physical activity (Sallis et al., 2016). Including these contextual details would strengthen the rationale for the study and further justify the focus on physical activity interventions for this population.
Secondly, it seems that a revision of the conclusions regarding the thematic analysis would help to enhance their clarity and impact. Specifically, the third theme, ‘to be the best’, may not be the most appropriate or sensitive descriptor within the context of a program designed for individuals with progressive cognitive impairments. The phrase appears to emphasize competition and superiority, which may be misaligned with the ethos of inclusivity, empowerment, and well-being that underpins dementia-friendly initiatives. A more suitable thematic title might be ‘to be better’, ‘to be improved’, or a similar term that captures the idea of personal fulfilment, participation, and progress without implying hierarchical achievement. This adjustment would better reflect the objectives of the intervention and the lived realities of its participants.
Sallis, J., Bull, F., Guthold, R., Heath, G., Inoue, S., Kelly, P., Oyeyemi, A., Perez, L., Richards, J. & Hallal, P. (2016) Progress in physical activity over the Olympic quadrennium. The Lancet, Physical Activity 2016 Series. http://dx.doi.org/10.1016/S0140-6736(16)30581-5.
World Health Organization. (2023a). Dementia. https://www.who.int/news-room/fact-sheets/detail/dementia.
Author Response
We would like to thank the reviewer for taking time to read our proposed manuscript and providing insightful and detailed feedback. We welcome the suggestions to improve our manuscript and contribution to this important area of research. Below are the responses to each comment.
Comment 1: This study makes a valuable and timely contribution to an under-researched area, particularly given the considerable challenges associated with promoting physical activity among individuals living with dementia. The proposed intervention of walking football is both creative and socially meaningful.
Response1: We would like to thank the reviewer for the positive feedback. We are encouraged that the reviewer considers our contribution to this field both timely and valuable; and recognises the value of the dementia friendly walking football intervention.
Comment 2: First, the introduction would benefit from further elaboration on the magnitude and gravity of the issue being addressed. Specifically, it would be helpful to foreground the scale of dementia as a global public health concern:
Currently, over 55 million people worldwide live with dementia, with more than 60% residing in low- and middle-income countries. Each year, approximately 10 million new cases are identified. Dementia is caused by a range of diseases and injuries affecting the brain, with Alzheimer’s disease representing the most prevalent form, responsible for 60–70% of cases. Dementia is now the seventh leading cause of death globally and a major cause of disability and dependency among older adults. In economic terms, the global cost of dementia was estimated at $1.3 trillion in 2019, with 50% of this burden attributable to informal care-giving, typically provided by family members who dedicate, on average, five hours per day to care and supervision (WHO, 2023).
In addition, physical inactivity is recognized as a significant modifiable risk factor for dementia. It has been estimated that approximately 3.8% of global dementia cases are attributable to insufficient physical activity (Sallis et al., 2016). Including these contextual details would strengthen the rationale for the study and further justify the focus on physical activity interventions for this population.
Response 2: We thank the reviewer for suggesting that we foreground the scale of dementia as a public health concern and the fact that physical inactivity is a significant modifiable risk factor for dementia. We also thank you for kindly including important relevant sources and references to support this. We have edited the text to make both issues more prominent within the manuscript. The new text has been inserted in at the beginning of the manuscript and referenced accordingly. Please see lines 31 – 37.
Comment 3: Secondly, it seems that a revision of the conclusions regarding the thematic analysis would help to enhance their clarity and impact. Specifically, the third theme, ‘to be the best’, may not be the most appropriate or sensitive descriptor within the context of a program designed for individuals with progressive cognitive impairments. The phrase appears to emphasize competition and superiority, which may be misaligned with the ethos of inclusivity, empowerment, and well-being that underpins dementia-friendly initiatives. A more suitable thematic title might be ‘to be better’, ‘to be improved’, or a similar term that captures the idea of personal fulfilment, participation, and progress without implying hierarchical achievement. This adjustment would better reflect the objectives of the intervention and the lived realities of its participants.
Response 3: Thank you for your reflections on our thematic analysis of the data and subsequent labelling of key themes. We appreciate your insights into the appropriateness of the phrase “To be the best” and take on board that this phrase may be interpreted as undermining to inclusivity and empowerment for people living with dementia and their families. As researchers, our aim is to promote and champion inclusivity and empowerment for people living with dementia. We also appreciate your suggested alternatives. In light of this, we have amended the name of this them to ‘a game changer’. As this phrase encompasses positive changes and impact, and is also derived from common use sports terminology. We hope this better reflects a flexible and inclusive approach rather than suggesting superiority and hierarchy. We have made edits to this in the abstract (line 23, Section 2 – Materials and methods (line 161), table 1, Findings (line 365 and lines 369 - 371), and discussion (line 461).
Reviewer 2 Report
Comments and Suggestions for Authors
Dear Authors
This study examined to a multi-stakeholder evaluation of a walking football group for people with dementia developed in partnership with a Premier League club. This study has been really well designed and written. It is interesting and excellent study and it presents good data on this important topic. I believe this is good issue in field of sports science section.
Minor concerns
Line 5: ‘Robinson 3,’ to ‘Robinson 3’
Line 5: ‘Killen 2 and’ to ‘Killen 2, and’
Line 7 and 9: ‘United Kingdom’ to ‘United Kingdom;’
Abstract
The original research articles contain the following headings: Background/Objectives, Methods, Results, and Conclusions.
Introduction
Line 36: ‘impairment.[4].’ to ‘impairment [4].’
Line 80: ‘MacRae et al.(2022) focused on the social impact alongside best practice recommendations [26]’ to ‘MacRae et al. [26] focused on the social impact alongside best practice recommendations’
Line 81: ‘Carone et al. (2016), explored the impact of a physical activity and sports based (including but not exclusively football) initiative for people living with young onset dementia and their families through a foot-ball club foundation [27].’ to ‘Carone et al. [27], explored the impact of a physical activity and sports based (including but not exclusively football) initiative for people living with young onset dementia and their families through a foot-ball club foundation.’
Line 108: ‘The aim of this paper’ to ‘The aim of this study’
I strongly recommend this manuscript should be edited by an English professional editor for more readable.
Methods
Line 127: ‘Mental Capacity Act (2005), Code of Practice [47].’ to ‘Mental Capacity Act [47], Code of Practice.
Line 128: Focus group consists of just n=4 and n=6. I think it has too small numbers. For this reason, you should insert this limitation in Limitation section.
Result
Line 201: NHS services to National Health Service (NHS) services. Abbreviations should be defined in the first instance.
Author Response
We would like to thank the reviewer for taking time to read our proposed manuscript and providing insightful and detailed comments to both quality and accuracy.
Comment 1: This study examined to a multi-stakeholder evaluation of a walking football group for people with dementia developed in partnership with a Premier League club. This study has been really well designed and written. It is interesting and excellent study and it presents good data on this important topic. I believe this is good issue in field of sports science section.
Response 1: We thank the reviewers for the positive comments on our design and report of our research. We are heartened that you find the study interesting and the topic of significant importance.
Comment 2: Line 5: ‘Robinson 3,’ to ‘Robinson 3’
Line 5: ‘Killen 2 and’ to ‘Killen 2, and’
Line 7 and 9: ‘United Kingdom’ to ‘United Kingdom;’
Response 2: Thank you for highlighting these punctuation errors. These have now been amended as suggested in lines 5,6,7 and 9 with a comma after Killen, the comma removed after Robinson, and semi colons added after the words United Kingdom.
Comment 3: Line 36: ‘impairment.[4].’ to ‘impairment [4].’
Response 3: Thank you for highlighting this punctuation error. The full stop has now been deleted (now line 42). Please also note this is now reference [7] due to additional edits as recommended by Reviewer 1.
Comment 4: Line 80: ‘MacRae et al.(2022) focused on the social impact alongside best practice recommendations [26]’ to ‘MacRae et al. [26] focused on the social impact alongside best practice recommendations’
Response 4: Thank you for identifying this referencing error. This has now been amended (now line 86). Please also note that this is now reference 29.
Comment 5: Line 81: ‘Carone et al. (2016), explored the impact of a physical activity and sports based (including but not exclusively football) initiative for people living with young onset dementia and their families through a foot-ball club foundation [27].’ to ‘Carone et al. [27], explored the impact of a physical activity and sports based (including but not exclusively football) initiative for people living with young onset dementia and their families through a foot-ball club foundation.’
Response 5: Thank you for identifying this referencing error. This has now been amended (now line 87). Please also note that this is now reference 30.
Comment 6: Line 108: ‘The aim of this paper’ to ‘The aim of this study’
Response 6: Thank you for suggesting this wording, this is more appropriate and has been amended accordingly. (Now line 114)
Comment 7: I strongly recommend this manuscript should be edited by an English professional editor for more readable.
Response 7: We apologise for the typographical errors and thank you for this suggestion. Unfortunately, we do not have sufficient additional funds to purchase this service. We have re-read the article, and made the recommended changes along with some additional small grammar, vocabulary and punctuation edits (marked in red text). We hope this will enhance the quality and accuracy of the manuscript sufficiently.
Comment 8: Line 127: ‘Mental Capacity Act (2005), Code of Practice [47].’ to ‘Mental Capacity Act [47], Code of Practice.
Response 8: Thank you for highlighting this. We have kept the year in the title as this is part of the title of the document but removed the parentheses around 2005 (now line 133).
Comment 9: Line 128: Focus group consists of just n=4 and n=6. I think it has too small numbers. For this reason, you should insert this limitation in Limitation section.
Response 9: Thank you for raising this. We agree that the small sample size is a limitation and as such have added the following text in the discussion section: We recognise that the findings of this research are limited to the views and experiences of a small number of service users and providers. This is reflective of the relatively small numbers of service users who attended at that time although the whole staff team delivering the service participated in their focus group. However, since then, the number of service users has since grown considerably, and the team has expanded. As such, we would like to engage with larger numbers of participants in the future to ensure that a broader range of perspectives and experiences are considered, facilitating ongoing reflection and change. (Lines 471 – 478).
Comment 10: Line 201: NHS services to National Health Service (NHS) services. Abbreviations should be defined in the first instance.
Response 10: We apologise for not spelling this out in the first instance. This has now been addressed in the text (line 208).